# A Person-Centered Approach to Moralization—The Case of Vaping

**DOI:** 10.3390/ijerph19095628

**Published:** 2022-05-05

**Authors:** Laura Arhiri, Mihaela A. Gherman, Andrei C. Holman

**Affiliations:** Faculty of Psychology and Education Sciences, Alexandru Ioan Cuza University, Str. Toma Cozma 3, 700554 Iasi, Romania; laura.arhiri@uaic.ro (L.A.); alexandra.gherman@uaic.ro (M.A.G.)

**Keywords:** attitude moralization, moral convictions, vaping, e-cigarettes, moral piggybacking, emotions, negative prototypes, smoking, disgust, self-control

## Abstract

Using e-cigarettes for smoking cessation is a controversial topic among health experts. Evidence suggests that vaping might have been moralized among the general public. Despite the detrimental consequences of moralizing health behaviors on social cohesion and health, some argue for using moralization strategically to prevent and combat vaping. We aim to add to the body of literature showing the dangers of moralization in health by proposing a person-centered approach to the moralization of anti-vaping attitudes. Our cross-sectional survey explores the moralization of anti-vaping attitudes and its predictors on a convenience sample of 348 Romanian never-vapers, before the final vote to severely restrict vaping. By fitting a hierarchical regression model on our data, we found support for a unique contribution of negative prototypes (β = 0.13) and opinions of vapers (β = 0.08) in predicting moralization, with significant contributions of piggybacking on moralized self-control, on moralized attitudes toward smoking and on sanctity/degradation, disgust, anger, harm to children, and gender. Together, these variables explained 56% of the variance of the moralization of anti-vaping attitudes. Our findings add to our knowledge of motivated moralization and advise against using moralization in health, suggesting that people may weaponize it to legitimize group dislike.

## 1. Introduction

How to regulate e-cigarettes and whether they should be used as cessation aids has been a topic of controversy among experts [1], with some supporting e-cigarette use for harm reduction/cessation [2] while others oppose it based on the lack of long-term health data, along with concerns about increased prevalence of nicotine addiction and re-normalization of nicotine consumption among youths who may not have otherwise taken up smoking [3]. The different positions taken by experts on the use and regulation of e-cigarettes may have influenced governmental officials from different countries to adopt diverse national regulation policies [4]: Countries such as Mexico have banned their distribution, exhibition, promotion, or manufacture, whereas countries such as the United Kingdom have regulated them far less restrictively than other nicotine products, recommending them for quitting smoking [5]. National health policies on lifestyle-related health behaviors such as vaping may contribute to attitude moralization in the general public [6,7,8]. This may have already occurred in contexts where vaping is seen as either *moral,* because it is perceived to help smokers quit [9,10], or *immoral*, where it is perceived as a gateway drug to smoking [10]. Thus, both pro-vaping and anti-vaping attitudes may have become moralized.

Attitude moralization in health severely affects social cohesion, leading to stigmatization, ostracism, and discrimination [7,8]. It also leads to reactance to the prescribed norm, with people eating more unhealthy snacks after being exposed to body-weight moralization (i.e., “obesity is immoral”) [11]. When vaping is considered moral because it is perceived as healthier than smoking (as opposed to when people perceive vaping as healthier than smoking without moral implications), they may erroneously conclude that vaping is simply healthy *because of the moral implication*, leading to diminished risk perception [9]. Despite these dangerous effects of moralizing health behaviors, recent studies recommend to policymakers’ appeal to moral values in anti-vaping health campaigns [9,10].

The main purpose of our study is to explore another dangerous implication of moralizing health behaviors: people might use moralization to legitimize their negative views of groups they dislike. Thus, we expected negative opinions and prototypes of vapers to significantly predict the moralization of anti-vaping attitudes, relying on the moral relevance ascribed to it in official public discourses. This way, people could feel morally justified and socially legitimized to act on their dislike openly, and stigmatize, ostracize, or discriminate against vapers [12]. People often use moralization to serve self-interested goals (i.e., *motivated moralization*) [13]. We explored this proposal, along with other predictors of anti-vaping attitude moralization in Romania, before the final vote on a controversial bill to restrict vaping. We also investigated several tracks of moral piggybacking (such as self-control or moral foundations). Throughout the manuscript, we will refer to “the moralization of anti-vaping attitudes” as “the moralization of vaping”, for an easier read.

### 1.1. Attitude Moralization—Predictors and Moral Piggybacking

*Attitude moralization* is the process through which moral relevance is attached to attitudes, transforming preferences into norms (i.e., moral mandates/convictions). Moralized attitudes are perceived as factually true and compulsory [6,14,15]. Moralization targets behaviors or practices perceived as harmful, especially when they hurt vulnerable individuals, such as children [6]. The strongest predictors of moralization are moral emotions and engaging in moral piggybacking [14,16,17,18,19,20]. Disgust protects the individual from contamination [6,19,20], and may be significantly associated with the moralization of vaping. Anger contributes to moralization when harm concerns are involved [14,21], which is why it may also be involved in the moralization of vaping.

Moral piggybacking is the cognitive mechanism of connecting an issue with existing moral principles or previously moralized objects based on similarities [6,14]. Cigarette smoking has been moralized since the 1990s [7,17]. This could constitute a moral piggybacking source for the moralization of vaping, given the similarities between e-cigarettes and tobacco cigarettes [22]. The moral foundations of harm/care and sanctity/degradation could be piggybacking tracks for moralizing vaping [3,10,23,24]. Harm/care describes our disliking and empathizing with harm incurred to others, while sanctity/degradation describes our striving to overcome the more instinctual aspects of our nature by avoiding physical and socio-psycho-behavioral contaminants [23]. Finally, vaping moralization could have been piggybacked on self-control: people believing that self-restraint is a moral value [25] may regard quitting smoking with the help of vaping as a moral failure.

### 1.2. A Person-Centered Approach to Attitude Moralization

Moralization has been studied from an act-based theoretical perspective, in line with the dominant ethical approaches in moral psychology (i.e., deontology and consequentialism). Thus, its known predictors characterize the practice/behavior rather than the human agents who set them in motion [6,14,16]. In this perspective, the primary function of moralization is protecting people from harmful or contaminating practices/entities, such as smoking [17] and eating meat [14]. However, considerations about harm (to children) are one of the strongest factors pushing moralization forward [17,26]. This shifts the focus from smoking as a practice to smokers as harm-doers [6]. Vapers were perceived to be morally flawed and stereotyped negatively before concerns of adverse health events (i.e., harm preoccupations) became popularized [27,28,29]. Moralization then serves to protect the innocent from harm-doers, in line with a person-centered, virtue ethics perspective which focuses on the basic human motivation to protect ourselves from dangerous others [30].

### 1.3. The Moralization of Health Behaviors

Health is a field which inherently invites moralization by operating the distinctions between “right”/“wrong” [6]. Since many of the behaviors previously considered lifestyle preferences were found responsible for chronic conditions, some were invested with moral relevance; being sick gradually became a moral failure in exercising sufficient self-control [6,7,17,26]. This happened for smoking in the USA [17] and is still occurring for obesity and smoking today in other contexts [7]. This can severely alter intergroup dynamics and social cohesion, because moralization leads to intolerance, discrimination, stigmatization, ostracism, and even violence toward the moral outgroup [8,15]. Hence, health and vaping should not be understood through the lens of morality, but through the lens of science [6].

### 1.4. Vaping in Romania

Currently, vaping is regulated by law in Romania more loosely than tobacco cigarettes. While advertising is prohibited and package health warnings are compulsory, purchase and consumption in public spaces are legal for those over 18 [31]. However, from 2018 to June 2021, the public discourse of top health officials relied on a strong anti-vaping rhetoric, invoking unsubstantiated scientific data according to which vaping poses more health risks than smoking [32]. This anti-vaping official rhetoric supported a bill categorizing vaping as a form of smoking to justify tighter regulations, proposed in 2018 by a group of politicians and the Ministry of Health. This categorization is improper, because e-cigarettes are devices which produce vapor, not smoke, and no scientific publications were cited in support of vaping being more dangerous than smoking, with most studies at the time suggesting otherwise [33,34]. While the bill was rejected in June 2021, prominent Romanian health officials and organizations publicly stated that a full nationwide ban on e-cigarettes is the morally correct action, for the good of the children [32,35]. This resembles the public rhetoric employed elsewhere which contributed health behavior moralization [7,8], as it invoked potential harm to children. Hence, it would be reasonable for us to infer that it may have laid the foundation for the moralization of vaping among the general public, and consequently conduct the present study.

### 1.5. The Current Study

We explored a person-centered approach to moralization by investigating whether negative perceptions of attitudinally dissimilar groups can predict vaping moralization [30]. We measured negative group perceptions as negative prototypes and negative opinions about attitudinally dissimilar others [27,28,29]. We also explored the moral piggybacking tracks contributing to vaping moralization: smoking, self-control, sanctity/degradation and harm/care. Finally, we expected vaping to be considered immoral based on contamination preoccupations and subsequent differences in the moral emotions [6,9,10,11,14,15,16,17,29,36,37], with disgust being a stronger predictor than anger because of its closer links with sanctity/degradation [23].

## 2. Materials and Methods

### 2.1. Participants and Procedure

A convenience sample of 348 individuals from Romania selected through snowballing techniques participated in our study. The inclusion criteria were not having vaped habitually, measured through self-report [38]; the 124 participants who reported having vaped more than twice were thus excluded. The study was conducted online, and all participants provided informed consent. They were rewarded two cash prizes for participation following a random draw. The study was conducted from January to May 2021, before the anti-vaping bill was voted in the Chamber of Deputies. The research received ethical approval from the Research Ethics Committee at the Faculty to which the authors are affiliated.

### 2.2. Instruments

*Socio-demographic characteristics.* Based on previous studies on moralization, participants were asked about gender, age, education, political orientation, and the importance of religion [25]. Political orientation was assessed according to [25], with the item: “In terms of politics, people usually talk about ‘left’ and ‘right’. Generally speaking, where would you place yourself on the following scale?” The item was followed by a 10-point Likert-type scale, ranging from 1—Left to 10—Right. Answers were categorized to reflect three political orientations: “Left” (1–3 scores); “Center” (4–7 scores); and “Right” (8–10 scores) [25]. The importance participants placed on religion was measured with one item used by [25] to assess the importance of religion to their lives: “How important is religion to your life?” Answers ranged from 1—not at all to 9—extremely important. They were categorized as “low religiousness” (1–3 scores), “medium religiousness” (4–6), and “high religiousness” (7–9). [26]. Past and present cigarette smoking was assessed with two items, on three-point Likert-type scales ranging from “never” to “daily”, with “occasionally” as the middle point [38].

*Moralization of vaping* and *cigarette smoking* were measured separately using the same scale that was adapted from studies on the moralization of other behaviors [14,15,18]. The scale comprises five items answered on a scale from 1 (not at all) to 5 (very much). The first two items were modeled on the ones used by [18]: “To what extent is your position on vaping/cigarette smoking a reflection of your core moral beliefs and convictions?”, “To what extent are your feelings about vaping/cigarette smoking deeply connected to your beliefs about ‘right’ and ‘wrong’?” [14], while the following two items were created by [14] to ensure face validity: “To what extent do you feel the issue of vaping/cigarette smoking is a moral issue (An issue where your attitude is based on moral values)?”, “When thinking about vaping/cigarette smoking, to what extent do you ‘just know’ that it is wrong?”. The fifth item was also added by [14] to explicitly assess moral content: “Overall, how much do you believe vaping/cigarette smoking is immoral?”. For both scales, individual scores were summed up, potentially ranging from 5 to 25, with higher total scores showing a greater moral relevance attached to attitudes toward vaping and, respectively, smoking. The validity of the scale has been confirmed in previous studies [14,15,18]. The internal consistency of the scales was adequate, with Cronbach’s α coefficients exceeding 0.7 (Table 1).

*Piggybacking* on the moral foundations of *sanctity/degradation* and *harm/care* was assessed with one item each, adapted from [14,39] and measured on a six-point scale whether the moralization of vaping occurred by appealing to concerns about the harmfulness of vaping (i.e., “Vaping is immoral because it harms the person and/or the others”) and about tainting vapers’ bodily and spiritual sanctity (i.e., “Vaping is immoral because it is a disgusting act which degrades the body and/or taints the soul”). Higher scores showed a higher degree of piggybacking on the respective moral foundation [23].

*The moralization of self-control* was assessed with an item adapted from [25], asking participants to what extent they considered “self-control and self-restraint” morally relevant on a six-point scale, ranging from 0—not morally relevant at all to 6—extremely morally relevant. Face validity was established by [25].

*Beliefs* about the harmful effects vaping might have on children, which could push the process of moralization, were assessed according to [14], a reliable measure. We constructed three items measured on a five-point Likert-type scale to evaluate whether vaping could harm children by exposing them to toxic substances; operating as a gateway drug for other nicotine products; or increasing the attractiveness of the latter via their flavored e-liquids [24,27,28,29]. Potential scores ranged from 3 to 15, and higher scores indicated stronger beliefs that vaping is harmful to children.

The scales for positive and negative prototypes of vapers and the scale for assessing opinions of vapers were adapted versions of the Prototypes of Tobacco Users Scale (POTUS), with good validity and reliability [29,40]. *Negative and positive prototypes of vapers* assessed to which extent a prototypical vaper is characterized by six positive (i.e., Sexy, Cool, Clean, Smart, Healthy, Attractive) and six negative traits (i.e., Trashy, Immature, Disgusting, Inconsiderate, Impolite, Arrogant), on a scale from 1 to 5 [29]. The scale was chosen because the negative traits have moral connotations [29]. Individual scores for positive and negative traits were summed up separately (potential range: 6–30), with higher total scores showing a more positive/negative prototype of vapers.

*The opinion about vapers* was measured with three items asking participants to evaluate whether this opinion is rather positive (1) or rather negative (5) and, respectively, how negative their negative opinions were (from 1—not at all negative to 4—extremely negative), and how positive their positive opinions were (from 1—not at all positive to 4 extremely positive) [29]. After reversing the scores for the item assessing how positive their positive opinions were, all individual scores were summed up. The higher the total score, the more negative the opinion about vapers, within a potential range from 3 to 15.

*Moral emotions*. Anger and disgust were measured with one item each, a valid and reliable measure, since people can easily identify their emotions [6,14,15,16,17,18,19,20]. Participants were asked to what extent they felt anger and, respectively, disgust at the thought of vaping, answering on a scale from 1—not at all to 5—very much.

### 2.3. Data Analysis Strategy

To test our research hypotheses, we conducted a hierarchical multiple regression in Jamovi 1.6.23.0 (The Jamovi Project, Sidney, Australia), to see whether introducing our hypothesized additional predictors would operate a significant change in the explained variance of moralization. This procedure allowed us to control for the influence of known predictors of moralization and to gain insight into our hypothesized relationships. Given the fact that our study was both confirmatory and exploratory, significance was assessed based on confidence intervals and *p*-values, at an alpha of 0.05. Because some predictors were correlated (Table 1), we analyzed the variance inflation factors to control for collinearity [41]. The highest value observed was 2.96 for negative emotions, which showed a negligible collinearity issue in our data.

## 3. Results

### 3.1. Participants’ Characteristics and Differences between Them According to Moralization of Vaping

Participants’ ages ranged from 18 to 58 (M = 25.9 ± 7.48), of whom 191 identified as male and 157 as female. Concerning their education, 216 reported having completed high school, 109—bachelor’s studies, and 23—master’s studies. Almost half were current or past smokers, with just over a quarter currently not smoking cigarettes. In terms of political orientation, most participants reported an affiliation to the right of the political spectrum. Religion was highly important for almost half of our sample (Table 2).

Significant differences in moralization of vaping were found according to *gender*, *age*, *education, current smoking status*, and *political orientation*, as illustrated by the results of Welch’s *t*- and *F*-tests presented in Table 2. Thus, our youngest participants moralized vaping significantly less than the ones aged 26 to 30, participants with high school studies moralized more than the ones with a master’s degree, and occasional smokers moralized more than habitual ones. Political centrists moralized less than people with political orientations to the right. Males moralized vaping more than females (Table 2).

### 3.2. Correlation Analyses

The higher the moralization of vaping, the higher the anger and disgust, beliefs about vaping harming children, piggybacking on purity and harm, the moralization of self-control and smoking, and the more negative the prototypes and opinions of vapers. In contrast, the higher the moralization of vaping, the less positive the prototype of vapers (Table 1).

### 3.3. Predictors of the Moralization of Vaping

A hierarchical regression analysis was conducted to investigate to what extent the moralization of vaping was predicted by opinions and, respectively, positive and negative prototypes of vapers, after controlling for the influence of socio-demographic characteristics (age, gender, studies, political orientation, past and present smoking status, religiousness), moral emotions of anger and disgust, beliefs about vaping harming children, moral piggybacking on the foundations of purity and harm, on the moralization of self-control and moralization of smoking (Table 3). The first model, comprising the aforementioned controlled variables, accounted for 55.6% variance, while the second model accounted for 57.6% variance, with a significant increase between the two of 2.3% variance.

In the first model, we found significant positive relationships between moralization of vaping and gender, disgust toward this practice, beliefs about vaping harming children, and moral piggybacking on sanctity/degradation on moral convictions about smoking and on moral convictions about self-control (Table 3). Thus, the moralization of vaping was piggybacked on the moral foundation of sanctity/degradation and on the moralization of smoking and self-control, while being facilitated by disgust and beliefs about harming the young. In addition, men moralized vaping more than women. No significant relationships were found between the moralization of vaping and participants’ age, studies, political orientation, religiousness, past and present smoking status, anger, and, respectively, moral piggybacking on harm.

In the second model, both anger and disgust significantly predicted the moralization of vaping (Table 3). Gender, beliefs about vaping harming children, and moral piggybacking on sanctity/degradation, the moralization of smoking, and self-control remained significant predictors of vaping moralization. We also found that the moralization of vaping was positively predicted by negative prototypes and opinions of vapers. No significant relationships were found for positive prototypes of vapers. Thus, our assumptions were partially confirmed.

## 4. Discussion

We examined whether attitudes toward vaping were moralized and evaluated the merits of a person-centered approach in predicting moralization. We also investigated the contribution of several tracks of piggybacking (i.e., moral foundations, the moralization of smoking, and the moralization of self-control). We found that negative prototypes and opinions about vapers significantly predicted the moralization of anti-vaping attitudes. Our results showed significant relationships for the following moral piggybacking routes: self-control moralization, the moralization of smoking, and sanctity/degradation. Finally, we found greater associations between the moral emotion of disgust as compared to anger, and significant associations between moralization and beliefs about harm to children.

Currently, the research on attitude moralization identified act-based predictors of this process: emotions toward the attitudinal object, its potential to bring about harmful consequences, moral piggybacking on previously moralized objects, and moral foundations [6]. For the first time, to our knowledge, our findings suggest that a person-centered approach to moralization may provide further insight into the process. People held negative opinions and prototypes of vapers prior to the emergence of serious health concerns about vaping [27,28]. Our results confirmed that they were uniquely, positively, and significantly associated with moralized anti-vaping attitudes. In line with past research on moral judgment, this shows that moral character evaluations may be associated with attitude moralization [30,42].

These results suggest that moralization could be used to legitimize dislike and disagreement. This could subsequently license acting upon them [15]. Moralization can enable actions otherwise perceived as socially unacceptable by granting people psychological standing and thus entitling them to act [12]. Having a negative *opinion* and *prototype* about a group of people is not sufficient to stigmatize or ostracize them, because these constructs are circumscribed to *preferences* [6]. However, if dislike of vapers was based on moral arguments, then any ensuing stigma or ostracism would likely be interpreted as moral activism, both socially acceptable and admirable [6,13]. Our research expands on the concept of motivated moralization, showing that it could occur not only to protect the self from an explicit external threat [13] or to legitimize action in the absence of moral interest [12], but also to protect and legitimize dislike toward a social group.

Positive prototypes of vapers, albeit negatively correlated with moralization, did not significantly predict it. This may be explained by the negativity bias in moral judgment, specifically *negativity dominance*, a phenomenon which describes how combining negative and positive traits yields assessments which are more negative than an algebraic sum of the valences [43].

Previous findings on the role of disgust and anger in triggering the process of moralization are mixed, with some studies supporting a unique role of disgust [19], some a combined contribution of both emotions [14], and others reporting no effect of either disgust or anger [20]. A recent perspective on these mixed results is that differences in the object of moralization could be responsible for this varied pattern of findings [20]. Our study contributes to this debate by suggesting that our participants associated both disgust and anger with the moralization of vaping. While the effect of anger should have been non-significant [24], the high contribution of the moralization of self-control to our model can help us understand this result. Self-control has been moralized to serve social group interests at the expense of individual interests [44]. Since vaping may have also been moralized in association with the perceived moral character of vapers, their perceived lack of self-control may have become a moral failure threatening social group interests. Anger is an emotion which invites action, as compared to disgust, which elicits avoidance or inhibition. Hence, it might be that, when moralization is connected by person-centered factors, anger responses play a role as well, targeting the perceived wrongdoers.

Regarding the moral piggybacking targets proposed, we found support for the moral foundation of sanctity/degradation, but not for harm/care. Our results are in line with recent findings, which show that evaluations of moral character based on norm violations are more severe when the violation depicts a bizarre act of impurity [42]. People tend to consider this type of act as particularly informative of moral character. Since past research found that vaping is seen as “strange” and “unusual” [27,28], this could explain why the moralization of vaping was associated with moral piggybacking on sanctity/degradation, especially from a person-centered perspective. The negligeable contribution of harm/care in explaining the moralization of vaping contrasts with the theoretical approaches which consider that violations of sanctity/degradation are circumscribed to harm/care transgressions [21], lending support to socio-intuitionist models, such as the moral foundations theory, which argue for a separation of the two values [23]. It seems that the perceived unnaturalness of vaping might weigh more than concerns about its potentially dangerous health effects in the process of attitude moralization, which would explain why the moralization of this practice was argued to have started before the advent of EVALI [24]. Future studies should verify these findings with experimental designs similarly to [14].

Our findings also supported the hypothesized moral piggybacking of the moralization of vaping through self-control. For individuals who moralize it, vaping may constitute a moral failure on the part of the vaper, similarly to obesity and other health issues [45]. Our results might suggest that the mechanism through which a health issue becomes a moral issue, shifting the dichotomy from healthy/sick to moral/immoral, could be the moralization of self-control [25]. This is in line with past research, which showed that lifestyle-related health conditions, such as substance abuse and obesity, are moralized when people are held responsible for choices perceived to be under their control [26].

Finally, the last significant piggybacking track for the moralization of vaping we found were moral convictions about smoking [17]. Our results contribute to the body of research showing the dangers of moralization in health [8]. In the past, moralization was employed strategically by public health officials in order to curtail smoking in the United States of America [6,17]. Although the rationale behind this strategy served the noble purpose of helping people to quit and deterring others from taking up smoking, the social effects of implementing it revealed that it backfired, because it gave people and institutions the license to disparage smokers [8]. The anger of the people at the realization of the association between smoking and lung cancer should have been pointed at the tobacco companies and other officials who were aware of this connection and chose to stay silent [6]. However, by moralizing the practice, smokers themselves ended up as targets of moral condemnation, stigma, and ostracism, as the public’s anger was channeled in their direction, based on the false assumption that addicted smokers have the freedom to choose to quit [6,7,8,15].

Our findings could also help to better explain the moralization of other health issues, such as obesity or being overweight. In this case, it would be difficult, if not impossible, to reduce the determinants of moralization to act-based factors studies so far, such as concerns about harm to others and contamination, since being overweight or obese is mainly detrimental to the individual’s health. A person-centered approach would better explain this process: body weight is linked to the individual’s moral character, being perceived as a consequence of preexisting laziness, gluttony, lack of self-control, and incompetence [26]. This can shed light on how the moralization of obesity can lead to discrimination and misattributions regarding the controllability of this affliction [45]. Future studies should investigate whether a person-centered approach to the moralization of obesity could better explain the strong moral position on this topic [45].

Moralization motivates and legitimizes action to such an extent that otherwise normal concerns, such as the welfare of the addicted, are suspended [6,21]. Given that it can be “weaponized” by officials and the public alike, it was argued that its intended use in public health and other social domains be discarded [8]. In support of this, it should also be considered how difficult it is to demoralize an attitude. According to [21], the only successful strategy employed to demoralize an object was to moralize its harmful effects, as happened when AIDS and homosexuality were demoralized by showing moralization’s devastating effects on the stigmatized. This would be difficult to do in the case of vaping, since not all vapers are former smokers trying to quit, so that stigmatizing them be considered immoral based on their suffering. Then, public health campaigns should underline the psychological suffering associated with the moralization of any health object, thus moralizing the judgment and condemnation of others. At any rate, in our view, a beneficial first step would be for health officials to stop using moralization themselves in an attempt to curtail health-risk behaviors. Considering that the decisions of policy makers may be informed by the general public’s position on the matter, non-governmental organizations and public figures should also contribute to the de-moralization of vaping. Beliefs and emotions functioned as deterrents of attitude moralization [14], and thus popularizing the psychological suffering of the groups targeted by moralization in media may be a way to educate beliefs and modify emotions [21]. For vaping specifically, medical information on addiction to smoking and the difficulties faced by people who are trying to quit by vaping could elicit emotions such as compassion and sympathy in the general public, thereby reducing disgust and anger [14]. Healthcare workers active on social media or who make television appearances could use these platforms to inform the public from a position of credibility, without having to defer to already formed opinions the public may hold.

### Limitations and Implications for Future Research

Given the limitations associated with the use of a cross-sectional study design, our findings should be further tested experimentally and longitudinally to confidently ascertain causation and directionality [10]. Since we cannot infer temporal causality based on our statistical analyses, some of our arguments from the discussion should be considered in this light and verified by future studies. The generalizability of our results may also have been limited by our sampling technique, as well as by the increased number of smokers among our participants, who represent over 70% of our participants, in a country where the prevalence of smoking is around 26% (WHO report on tobacco epidemic, 2019). However, this high proportion of smokers in our sample gives us insight into the group dynamics among smokers who did not (manage to) use e-cigarettes for quitting smoking. Our findings may suggest that their high moralization of self-control, the predictor which explained the most variance in the moralization of vaping, might have predisposed them to be less willing to use quitting smoking aids. This would be in line with the previous qualitative findings, which showed that using quitting aids such as e-cigarettes to quit smoking was sometimes perceived as a sign of weakness [15]. Because we did not measure their intentions to quit smoking or their attempts at using other quitting aids, this speculative conclusion should be verified by future studies.

Future research should also look into certain personality traits or individual dispositions which may increase the propensity to moralize based on negative opinions and prototypes. In our study, we did not find any socio-economic predictors relevant to this issue, aside from gender. However, other factors should be examined, such as low self-worth and low self-esteem. For instance, [12,13] showed that self-affirmation helps people moralize their questionable behaviors less, because feeling self-assured or self-secure undermined the need to self-enhance by appealing to moral values.

## 5. Conclusions

Our study showed that attitudes toward vaping have been invested with moral relevance based on negative emotions, dislike, and disagreement with vapers as a social group, moral concerns of contamination and unnaturalness, social pressures, and moral piggybacking on smoking and self-control. This revealed that negative opinions and prototypes of an outgroup could motivate moralization, while also showing previously unforeseen consequences of the moralization of smoking. Although smoking cigarettes has been invested with moral relevance more than twenty years ago, we are still facing its consequences today. For all these reasons, we conclude that moralization should not be used by public health officials as a strategy to curtail health-risk behaviors and should be discouraged among the public as well. The public should be educated about the difficulties in exercising self-control in addictions, such as nicotine addiction, and other lifestyle-related afflictions, such as obesity, so that moralization and its social consequences are less likely to occur. Such cognitively-oriented initiatives should be accompanied by emotionally oriented ones, aiming to sensitize the public to the moralized groups’ suffering.

## Figures and Tables

**Table 1 ijerph-19-05628-t001:** Means, standard deviations, Cronbach’s α (on the diagonal), and Pearson’s r correlations between moralization of vaping and its predictors.

	M ± SD	1		2		3		4		5		6		7		8	9	10		11
**1. Moralization of vaping**	14.9 ± 3.98	*0.843*																		
**2. Harm to children**	11.2 ± 2.83	0.495	***	*0.716*																
**3. Sanctity/degradation**	2.71 ± 1.48	0.421	***	0.299	***	-														
**4. Harm/care**	3.42 ± 1.74	0.226	***	0.215	***	0.485	***	-												
**5. Moralization of smoking**	14.3 ± 5.34	0.334	***	0.207	***	0.208	***	0.193	***	*0.882*										
**6. Moralization of** **self-control**	3.32 ± 1.41	0.59	***	0.291	***	0.231	***	0.154	**	0.148	**	-								
**7. Anger**	2.74 ± 1.38	0.245	***	0.026		0.054		0.028		0.076		0.315	***	-						
**8. Disgust**	2.82 ± 1.42	0.307	***	0.084		0.027		−0.079		0.102		0.342	***	0.167	**	-				
**9. Positive prototype**	11.3 ± 4.76	−0.173	**	−0.231	***	−0.057		−0.119	*	−0.079		−0.095		−0.107	*	0.004	*0.92*			
**10. Negative prototype**	11.5 ± 4.01	0.36	***	0.221	***	0.244	***	0.14	**	0.178	***	0.222	***	0.002		0.087	−0.04	*0.923*		
**11. Opinion of vapers**	8.19 ± 2.55	0.236	***	0.157	**	0.142	**	0.096		0.097		0.135	*	0.06		0.05	−0.097	0.156	**	*0.84*

* *p* < 0.05, ** *p* < 0.01, *** *p* < 0.001; Cronbach’s Alpha in italic.

**Table 2 ijerph-19-05628-t002:** Participants’ characteristics and differences in moralization of vaping.

Characteristic	Categories	Frequency & Percentage	M ^a^	SD ^a^	Welch’s *t* (*p*)	Welch’s *F* (*p*)	Games-Howell
Gender	Female	157 (45.1%)	13.6	4.7	5.42 (<0.001)	-	-
	Male	191 (54.9%)	15.9	2.88			
Age	18–20 (a)	95 (27.3%)	13.9	4.08	-	3.33 (0.013)	a < c **
	21–25 (b)	111 (31.9%)	15	4.63			
	26–30 (c)	70 (20.1%)	15.9	2.78			
	31–40 (d)	50 (14.4%)	14.9	3.43			
	41+ (e)	22 (6.3%)	14.7	3.81			
Studies	Highschool (a)	216 (62.1%)	15.4	3.7	-	7.89 (<0.001)	a > c **
	Bachelor’s (b)	109 (31.3%)	14.4	3.93			
	Master’s (c)	23 (6.6%)	11.7	5			
Current smoking status	Non-smoker (a)	92 (26.4%)	15.1	4.85	-	7.15 (<0.001)	b > c ***
	Occasional smoker (b)	107 (30.7%)	15.7	3.13			
	Daily smoker (c)	149 (42.8%)	14.1	3.81			
Past smoking status	Non-smoker	62 (17.8%)	15	5.08	-	2.45 (0.090)	-
	Occasional smoker	137 (39.4%)	15.3	3.46			
	Daily smoker	149 (42.8%)	14.4	3.88			
Political orientation	Left (a)	46 (13.2%)	14.8	3.66	-	4.42 (0.014)	b < c **
	Center (b)	147 (42.2%)	14.2	4.22			
	Right (c)	155 (44.5%)	15.5	3.73			
Religiousness	Low	94 (27%)	14.1	4.39	-	2.61 (0.076)	-
	Medium	99 (28.4%)	14.7	3.92			
	High	155 (44.5%)	15.4	3.7			

** *p* < 0.01, *** *p* < 0.001; ^a^ M and SD are the means and standard deviations for the total scores on the moralization of vaping scale.

**Table 3 ijerph-19-05628-t003:** Results of hierarchical regression analyses: Predictors of moralization of vaping.

	Model 1								Model 2							
			95% CI				95% CI				95% CI				95% CI	
	Estimate	SE	LL	UL	*p*	β	LL	UL	Estimate	SE	LL	UL	*p*	β	LL	UL
**Intercept**	4.09	1.2	1.73	6.45	<0.001				5.24	1.46	2.37	8.11	<0.001	0	0	0
**Age**	0.02	0.02	−0.02	0.06	0.411	0.03	−0.04	0.11	0.02	0.02	−0.02	0.05	0.435	0.03	−0.04	0.10
**Gender**	−0.82	0.32	−1.44	−0.20	0.01	−0.10	−0.18	−0.03	−0.80	0.31	−1.41	−0.19	0.01	−0.10	−0.18	−0.02
**Studies**	−0.41	0.25	−0.90	0.09	0.106	−0.06	−0.14	0.01	−0.32	0.25	−0.81	0.17	0.2	−0.05	−0.13	0.03
**Present smoking**	−0.22	0.29	−0.78	0.34	0.436	−0.05	−0.16	0.07	−0.14	0.28	−0.68	0.41	0.628	−0.03	−0.14	0.09
**Past smoking**	0.21	0.32	−0.41	0.83	0.504	0.04	−0.08	0.16	0.08	0.31	−0.54	0.69	0.808	0.01	−0.10	0.13
**Political orientation (left-right)**	0.05	0.08	−0.10	0.20	0.483	0.03	−0.05	0.10	0.04	0.07	−0.11	0.18	0.608	0.02	−0.05	0.09
**Religiousness**	−0.03	0.06	−0.14	0.09	0.654	−0.02	−0.09	0.60	−0.05	0.06	−0.16	0.06	0.393	−0.03	−0.11	0.04
**Harm to children**	0.37	0.06	0.26	0.48	<0.001	0.26	0.18	0.34	0.33	0.06	0.22	0.44	<0.001	0.23	0.15	0.31
**Sanctity/degradation**	0.61	0.12	0.38	0.83	<0.001	0.23	0.14	0.31	0.55	0.11	0.32	0.77	<0.001	0.20	0.12	0.29
**Harm/care**	−0.03	0.10	−0.22	0.17	0.789	−0.01	−0.10	0.07	−0.04	0.10	−0.23	0.15	0.691	−0.02	−0.10	0.07
**Moralization of smoking**	0.12	0.03	0.06	0.18	<0.001	0.20	0.09	0.24	0.11	0.03	0.05	0.16	<0.001	0.14	0.07	0.22
**Moralization of self-control**	0.95	0.13	0.70	1.2	<0.001	0.34	0.25	0.43	0.90	0.12	0.65	1.14	<0.001	0.318	0.232	0.41
**Anger**	0.21	0.11	−0.00	0.43	0.055	0.07	−0.00	0.15	0.22	0.11	0.01	0.43	0.045	0.08	0.00	0.15
**Disgust**	0.29	0.11	0.08	0.51	0.008	0.11	0.03	0.18	0.30	0.11	0.08	0.50	0.007	0.10	0.03	0.18
**Positive prototype**									−0.03	0.03	−0.09	0.03	0.277	−0.04	−0.11	0.03
**Negative prototype**									0.12	0.04	0.05	0.20	0.001	0.13	0.05	0.20
**Opinion typical vaper**									0.13	0.06	0.24	0.02	0.026	0.08	0.15	0.01
**Model fit**	R² = 0.574, Adjusted R² = 0.556, F(14, 333) = 32, *p* < 0.001								R² = 0.597, Adjusted R² = 0.576, F(17, 330) = 28.7, *p* < 0.001							
**Model comparison**	ΔR² = 0.023, F(3, 330) = 6.27, *p* < 0.001															

Estimate = Unstandardized Regression Coefficients; SE = Standard Errors; CI = Confidence Intervals; LL = Lower Limits of Confidence Intervals; UL = Upper Limits of Confidence Intervals; β = Standardized Regression Coefficients.

## Data Availability

The data presented in this study are available on request from the corresponding author. The data are not publicly available due to informed consent specifying that data will not be made publicly available and will only be visualized for research purposes.

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
