# Peer review of "A Person-Centered Approach to Moralization—The Case of Vaping"

_ijerph, 2022, doi:10.3390/ijerph19095628_

Round 1

Reviewer 1 Report

Abstract

The abstract should include more information about the methods (what variables were used, what stats) and should include numerical results.

Introduction

Line 29: The reason for comparing to COVID-19 is unclear and COVID is not related to this manuscript. I suggest removing this comparison.,

Line 30: I am not convinced that there is a sufficient body of literature to suggest that “e-cigarettes were shown to be particularly effective in reducing the health-related harms of tobacco smoking.” Are there epidemiological studies to confirm this statement?

Line 31-32: “as they address this type of addiction more comprehensively” – it is not clear to what this refers to; more comprehensively than what?

Line 45-46: The authors describe the process of moralization of vaping in Romania, but don’t provide support for this statement. It may benefit to give more background on Romanian tobacco policy and how it has become moralized.

Line 51: I also think the authors need to provide more definition of piggybacking.

Line 83-89: It’s not clear how this reference suggests the moralization of vaping. Can social stigma and negative stereotypes be negative perceptions that are not necessarily due to morals?

Throughout the introduction, the authors discuss harm-reduction and abstinence-only/zero tolerance as the only options without acknowledging a middle ground. For instance, the U.S. doesn’t really have abstinence-only policies, given that nicotine replacement therapies are an approved medication for smoking cessation. It is also possible to acknowledge that e-cigarettes may help smokers reduce harms and may be dangerous to children, and for policies in this gray area to exist (e.g., limited e-cigarette market with devices proven to protect public health, flavor/age restrictions to be less appealing to children).

Line 163-164: Again, the authors bring up zero-tolerance policies about smokers stopping without aids – can they give some examples? I’m sure policies exist but most countries I know of have some type of cessation aid (medication, counseling, etc.)

Section 1.3 can be more clear and straightforward. It is difficult to follow the authors’ planes for the paper.

Methods

Lines 185-187: The sample description and Table 1 should be included in the results rather than methods. It is hard to understand some of this information before the variables are introduced.

Table 1: Make more clear what M and SD refer to (I think this are moralization scores?). Also please clarify “occasional” versus “smoker” (does this mean daily). Those who smoke occasionally are still smokers.

Section 2.2 – more information is needed regarding the measurement of variables. How were political orientation and importance of religion measured?

The moralization of vaping scale would benefit by providing all 5 items in the text (or in supplementary material). Additionally, has this scale been assessed for validity? Do we know how well people can reflect on their own moralization of vaping? These should be discussed or mentioned as potential limitations. Finally, maximum and minimum possible scores should be provided. These comments, particularly validity, apply to all measurements used. The sanctity/degradation and harm/care can use a clearer explanation and definition, as I’m not sure I really understand them.

Section 2.3 – Much more information is needed here. For instance, none of the statistics run for Table 1 or Table 2 are described here. What was considered statistically significant? Additionally, did the authors consider running mediation models? They seem like they would be relevant for the research question.

Results

Section 3.3. IS it appropriate to use the words antecedents with cross-sectional data? We don’t know the temporality though antecedents implies this. Again, a mediation model may help the authors get closer to having temporal order.

Table 3. The abbreviations should be described in a footnote. Additionally, what is the difference between estimate and B? Finally, we suggest the authors round to 2 decimal places (the table is hard to read).

Discussion

The discussion is well-written and very descriptive, though also long and difficult to digest in one read (this also applies to the introduction to some extent). I suggest the authors try to highlight the key points more, abbreviate some areas, and write more directly and succinctly.

I suggest providing a brief summary of the key finding at the end of the first paragraph.

Line 326-327: “attitudes about practices may become moral convictions” – This statement implies causality that can not be concluded based on the cross-sectional data used.

Line 328-332: I suggest breaking this sentence into two.

Line 335-336: “vaping should not have been considered a transgression toward the self” – is it possible that people are just misinformed on the harms of vaping, rather than having moral convictions?

Line 343-344: Again, this suggests temporality that can not be inferred. It also isn’t clear what results are permitting the conclusion that people were acting upon these morals (behavior or support for policies were not measured)

Section 4.1: The authors should mention a potential limitation of generalizability.

Reviewer 2 Report

My reading of your paper is that it has two aims.  The first aim is to support the side of the contentious vaping debate that is opposed to restrictions on vaping; the second is to argue that vaping and vapers should not be stigmatised.  Pursuing the first aim involves a superficial analysis of the evidence for and against vaping and cherry-picking of arguments in favour of liberal vaping regulation.  You set up a ‘straw man’ argument of ‘zero-tolerance policies’ informed by ‘moralization’ that you subsequently attempt to confute.  However, you provide little or no evidence that ‘moralization’ currently plays a significant role in vaping health policy and your survey has little relevance to health policy on vaping.  If you want to make the case for liberal vaping regulation, you should write a paper that addresses this issue and objectively consider all the pros and cons.

However, pursuing the second aim has some merit and your discussion of ‘moralization ‘is interesting and could, in my view, be developed into a publishable paper.  Moralization of vaping could be extended to obesity, which you already mention, but also more broadly to anti-vaccination, drug addiction, mental health, poverty, and illnesses such as chronic fatigue syndrome, all of which involve stigmatisation of those concerned.  Your survey would have some relevance in such a paper, whereas the views of a convenience sample of Romanians to health policy on vaping have little or no general application. 

You use the term ‘zero-tolerance policy’ consistently without defining what you mean.  Is requiring a prescription for vaping, as required in Australia, a ‘zero-tolerance policy’?  What about only being able to buy the full range of vaping flavours from a specialist vape store as is the case in New Zealand?  The WHO report you quote does not recommend banning vaping entirely, which is what your narrative and the term ‘zero-tolerance’ implies.  WHO simply recommends that ENDS should be strictly regulated, which is arguably a reasonable response to the potential harm of vaping to young people.  You may disagree with this response, and people do, but to argue that it is ‘rooted in moral-emotional intuitions’ that ‘legitimise the use of any type of means to a desired end’, with no evidence to support this assertion, is the straw man argument I have mentioned.

Moralization may once have been a tobacco control strategy in the USA, but that is ancient history and I do not believe this is current policy anywhere.  On the contrary, campaigns such as the’ truth’ campaign in the US are designed to destigmatise smoking and place the blame for smoking where it belongs, on the tobacco industry.  On page 3, line 114, you say ‘The potential harm to children incurred by passive smoking led to a moral panic.’  This is a pejorative interpretation of legitimate concerns about the safety of children and illustrates the partisan view that permeates your analysis of the evidence you present.  On page 4, line 163, you say ‘zero-tolerance approaches to vaping imply that smokers should stop smoking without aids, which leaves them to rely on willpower and self-control’.  This may have been said in Romania, but generally the statement is incorrect; tobacco control policies typically include quit support measures and help for smokers who want to quit. 

There are other examples in your paper I could quote that make my point that your selective choice of evidence and debateable interpretation of its implications creates a misleading advocacy for a liberal approach to vaping rather than a balanced and objective account of the arguments.  If advocacy is your objective, I suggest that you concentrate on this objective rather than trying to combine in the same paper two ideas that are only tenuously related.

As far as your survey is concerned, there are some limitations you need to address.  Your sample is a convenience sample that is heavily biased towards smokers.  Only 26% of your respondents were not current or occasional smokers in a country where smoking prevalence is around 27%.  You need to consider how representative of the Romanian population the view of your respondents would be.  Your survey sample consisted of members of the general public; thus, you cannot legitimately ascribe their views to public health policy makers.  This is one reason why I believe you should separate your study and the moralistic views of the general public from the views and actions of policy makers and politicians. Your research provides no evidence of the views or actions of the latter.

A minor point. On page 4, line 185, you say the age range of your sample was 18 to 40, but on page 7, line 266 you say the participants were aged 18 to 58.

Round 2

Reviewer 1 Report

The paper, particularly the introduction, is inappropriately long. By my estimation, the introduction is almost 3500 words, which is longer than the maximum word length for full manuscripts for many journals. As a result, this paper is difficult to read and review and I’m concerned that length will limit the readership and possible impact. The authors need to be more concise in their introduction (and the full paper) to focus on what is essential to setting up the study – why is this topic important, what are the key facts that we know, and what are the key facts that are unknown. Please be succinct, and where possible write one general sentence that can be supported by several references. I suggest trying to reduce the words by at least 50% (probably more). Some potential suggestions are listed below (note these are examples, not an exhaustive list):

  • The first sentence on tobacco use deaths does not add value to the paper. I suggest cutting this sentence and start with discussion of the controversy about e-cigarettes.
  • I recommend cutting direct quotes from other papers. This is very uncommon in the scientific literature and does not add value, just length. Rather than providing full quotes from individual studies, synthesize that literature by making one claim with several references to support it. For instance, this paragraph can be summarized very briefly – “How to regulate e-cigarettes and whether they should be used as cessation aids has been a topic of controversy among experts [2], with some supporting e-cigarette use for harm reduction/cessation [4] while others oppose based on the lack of long-term health data [3].” The authors should make more attempts to cut down paragraphs into 1-2 sentence rather than extensively elaborating on every point.
  • Line 40: the authors state that the different positions taken by experts have led governments to adopt diverse regulations. Is there any demonstrated link between expert position and government regulations, or is this just an assumption by the authors?
  • Line 42: There is no need to state “according to the Institute…” – just provide the information and citation.
  • Though I asked for a definition of piggybacking, there are now 3-4 paragraphs on piggybacking which is far too much. Again, the authors need to be more concise with their approach and focus only on what is essential to justify their study. For instance, the authors state their believes on moralization of vaping (line 242) and then state a recent study confirms this. In this case, just state the scientific evidence and leave personal opinions out of the paper.
  • Every section is far too long. Rather than 1 page on attitude moralization, 1 page on person-centered approach, 1 page on moralization of health behaviors, the authors should shoot for 1-2 paragraphs on each of these topics.
  • I’m still not convinced by the authors claims that the health officials/organizations public statements laid the foundation for moralization among the general public – is there any data to support this? Could the opposite be true, where moralization among the public laid the foundation for health and government officials?
  • Section 1.5 is too long – the authors should be able to explain the current study in a few sentences, rather than a few paragraphs.

Methods: Again, this is a section with far too much detail and can be significantly shortened while still allowing the reader to understand and potentially replicate the methods. Please try to provide only necessary details and if necessary, provide extra detail in supplemental materials. Some potential suggestions are listed below (these are examples, not an exhaustive list)

  • In Section 2.2, rather than clarifying “for the purposes of assessing differences…” and just explain that scales from 1-9 were categorized as low (scores 1-3), medium (scores 4-6) or high (7-9). This comment generally applies to both political affiliation and religion.
  • In general, the authors again can work on conciseness. For example: “Moralization of vaping and cigarette smoking were measured separately using the same scale that was adapted from studies on moralization of other behaviors [15, 16, 19]. The scale comprises…” This significantly shortens the first sentence and reduces some sentences after the scale explanation. The several sentences after line 760 (“the validity of the scale…”) are not essential and can be removed. If the authors feel strongly about retaining this information, perhaps some details can be provided as supplemental materials. This applies to the Beliefs and POTUS as well.
  • The details on the digitized corpus of books written in English…are unnecessary. It is sufficient to state that face validity was established and provide the citation without the details of how this was done.
  • In general, the same comments apply from the Introduction. The study is well explained but the authors overwhelm readers with unnecessary details that make the manuscript very difficult to read.
  • Again, the authors should explain analyses from Table 2 (welch’s t, welch’s F, etc.) in the statistical analyses. All analyses that were conducted should be reported.

Discussion

  • Line 1403: “Unlike smoking, vaping should not be considered a moral transgression to such a high extent…” Are the authors stating that smoking should be considered a moral transgression?
  • Line 1409: A citation should be provided for “e-cigarettes having the potential to constitute a gateway drug for smoking.”
  • The authors should be cautious in lines 1484 – 1491 about better explaining the moralization of other health issues. Other health issues were not assessed, so how can you be sure these results generalize to other conditions?

Author Response

Please see the file attached.

Reviewer 2 Report

You have substantially re-oriented your original paper, largely eschewing your advocacy for liberal vaping laws, and concentrating on the implications of moralizing vapers and other socially stigmatised groups.  In my view, this re-orientation has produced a much more compelling ms with greater general interest.

Before you finalise the paper, you might liken to consider the following issues.

As you point out, the regulation of ENDS is controversial, with intransigent advocates at both ends of the liberalisation continuum and most public health advocates somewhere in between.  The problem is that there is no consensus on how best to achieve the public health goals of minimal smoking and protection of young people from harm (from tobacco or vaping).  Nevertheless, the stigmatisation of smokers, vapers, drug addicts, the morbidly obese, gamblers or anti-vaxxers is arguably cruel and likely to be unhelpful in their rehabilitation.  You have couched this stigmatisation problem in terms of policy makers moralization of vapers, but I think this perspective is too narrow.  Policy makers may be sympathetic to vapers and other stigmatised groups, but their decisions are inevitably constricted by public opinion.  For example, subsidised bariatric surgery may be a very sensible response to obesity, but if the public are strongly opposed to this move on the grounds that obesity is simply a matter of individual self-control, it would take a brave government to disregard this opposition.  You do mention the problem of public opinion, but I think you could emphasise this issue and perhaps suggest how it could be addressed, rather than attributing the burden of responsibility or moralization to policy makers alone.

On page 5, you quote the finding that vaping is 49% more effective than other NRTs and cite the Cochrane review that supports the superior efficacy of ENDS as a cessation device.  This narrative gives the impression that, if only smokers had unlimited access to ENDS, they would quit in droves. But the fact is that neither ENDS nor NRTs are particularly successful in achieving quitting, and the results cited were based on controlled trials involving cessation support, not unsupported, real-world experience. 

Also, the reality is more complex than simply arguing that ENDS help some smokers to quit so there should be few, if any, restrictions on their availability or use.  ENDS clearly help some smokers to quit, but not always, and there is some evidence that dual use, which sometimes happens, might be worse than either smoking or vaping on their own.  Furthermore, there is evidence of increasing vaping among young people; even if this practice is not a gateway to smoking, nicotine addiction is not desirable, and nicotine is not harmless to young people’s developing brains.  And, even if second-hand vapour is harmless, non-vapers presumably have their right not to breathe in clouds of Unicorn Milk vapour if they don’t want to.

I appreciate that your view on vaping puts you at the more liberal end of the vaping intervention continuum, but I think your case would be stronger if you presented a more balanced evaluation of the evidence for and against ENDS as a cessation device and conceded that e-cigarettes are not a silver bullet for smoking cessation.

I am not familiar with the vaping legislation debate in Romania so I cannot tell how accurate your account is of what transpired. However, statements such as “top health officials relied on a strong anti-vaping rhetoric, invoking non-existent scientific data according to which vaping poses more health risks than smoking” and “blatantly ignoring in the content of the bill the structural and functional differences between tobacco cigarettes and e-cigarettes, and falsely claiming a lack of scientific proof to show vaping is less harmful” could arguably be defamatory, even if they are true.  I would tone down the rhetoric of this section.  You could convey your disagreement with the views expressed without being quite so provocative and I believe this would enhance the credibility of your case.

In your conclusion you say: “For all these reasons, we conclude that moralization should not be used by public health officials as a strategy to curtail health-risk behaviors and should be discouraged among the public as well.”  Constraining the behaviour of public health officials is achievable, but I am not sure how you would propose discouraging the public.  It would be good if you could speculate on how this might be done.  

Your paper is generally well constructed and well written; however, the Introduction is very long and would benefit from some judicious editing to shorten it and tighten its focus.  

Page 16, line 1746: ‘who’ should be ‘which’.

Author Response

Please see the file attached.
